# Demoralization and Its Association with Quality of Life, Sleep Quality, Spiritual Interests, and Suicide Risk in Breast Cancer Inpatients: A Cross-Sectional Study

**DOI:** 10.3390/ijerph191912815

**Published:** 2022-10-06

**Authors:** Ting-Gang Chang, Chih-Chiang Hung, Pei-Ching Huang, Chiann-Yi Hsu, Ting-Ting Yen

**Affiliations:** 1Department of Psychiatry, Taichung Veterans General Hospital, Taichung 407612, Taiwan; 2School of Psychology, Chung Shan Medical University, Taichung 40201, Taiwan; 3School of Medicine, College of Medicine, National Yung Ming Chiao Tung University, Taipei 11221, Taiwan; 4Department of Post-Baccalaureate Medicine, College of Medicine, National Chung Hsing University, Taichung 40227, Taiwan; 5Division of Breast Surgery, Department of Surgery, Taichung Veterans General Hospital, Taichung 407612, Taiwan; 6Department of Applied Cosmetology, College of Human Science and Social Innovation, Hungkuang University, Taichung 433304, Taiwan; 7Cancer Prevention and Control Center, Taichung Veterans General Hospital, Taichung 407612, Taiwan; 8Biostatistics Task Force, Taichung Veterans General Hospital, Taichung 407612, Taiwan; 9Department of Otorhinolaryngology, Taichung Veterans General Hospital, Taichung 407612, Taiwan

**Keywords:** demoralization, breast cancer, quality of life, spiritual interests, suicide, depression

## Abstract

With decreasing mortality, the quality of life, spiritual needs, and mental health of breast cancer patients have become increasingly important. Demoralization is a poor prognostic factor for cancer patients. The extent of demoralization in breast cancer patients and its association with these factors remains unclear. This cross-sectional study was conducted at a Taiwanese medical center. We enrolled 121 participants (34 with high demoralization and 87 with low demoralization, as per the Mandarin Version of Demoralization Scale). High demoralization was associated with reduced quality of life, sleep quality, and spiritual interests. Multivariate analyses revealed that the scores of the European Organization for Research and Treatment of Cancer Quality of Life Questionnaire ≥ 62.5 (OR = 0.21, *p* = 0.002) and Spiritual Interests Related to Illness Tool Chinese Version ≥ 3.66 (OR = 0.11, *p* < 0.001) were associated with low demoralization. Demoralized patients with depression had a poorer quality of life and sleep quality. Although not statistically significant, depressed and demoralized participants were at a higher risk of suicide. Cancer patients with both depression and demoralization had the worst prognosis. Breast cancer patients exhibited demoralization when they had unmet bio-psycho-social-spiritual needs. An early assessment of demoralization may improve holistic healthcare for breast cancer patients.

## 1. Introduction

Breast cancer accounts for 11.7% of all new cancer cases [1]. Female breast cancer has surpassed lung cancer as the most commonly diagnosed cancer, with an estimated 2.3 million new cases (11.7%) and a 6.9% mortality rate [2]. By 2040, the burden of breast cancer is predicted to increase to over 3 million new cases and 1 million deaths every year owing to population growth and aging alone [3]. In 2019, data from the Taiwan Cancer Registry Center showed that breast cancer is fourth in the whole cause of cancer mortality, and the second leading cause of cancer death in women. Breast cancer is also the most common cancer among women and the whole population [4]. The literature shows that breast cancer screening and adjuvant therapy have important roles in reducing breast cancer mortality [5,6]. Research shows that multidisciplinary teams can help reduce breast cancer mortality by integrating surgery, radiation, and medical oncology [7]. Breast cancer mortality rates declined between 1975 and 2011 [8]. With this decreasing mortality, the quality of life, spiritual needs, and mental health [9] of breast cancer patients have become increasingly important. However, aside from the biological aspect of breast cancer, the psychological and spiritual needs of patients with breast cancer are often not met, and related research is lacking.

### 1.1. Demoralization and Breast Cancer

Demoralization is a normal psychological response to painful, advanced, terminal diseases [10,11,12,13], and/or heroin addiction [14]. Demoralization includes symptoms in five domains: loss of meaning, dysphoria, disheartenment, helplessness, and sense of failure, which can be assessed using rating scales [15,16]. The specific manifestation of demoralization differs from the American Psychiatric Association’s diagnostic criteria for major depressive disorder. In the diagnosis, there is a focus on depression and anhedonia. Currently, demoralization does not refer to a specific brain pathology, and the understanding of demoralization is not comprehensive. Fang et al. reported that 42% (n = 200) of cancer patients in Taiwan were highly demoralized [17]. Nevertheless, the incidence and severity of different types of cancer remain unclear. Previous studies on demoralization and breast cancer treatment strategies have been limited to countries such as Taiwan, China, Turkey, and Lebanon [18,19,20,21]. They have found that demoralization in breast cancer is related to breast and hair loss [20,21]. Triple-negative breast cancer (TNBC) is a poor prognostic factor for breast cancer [22]. Patients with breast cancer and TNBC may have a higher risk of demoralization. However, the association between demoralization and TNBC is not well understood.

### 1.2. Demoralization and Its Association with Quality of Life, Sleep Quality, Spiritual Interests and Suicide

Quality of life is an important factor that may have a bi-directional relationship with demoralization in cancer patients [23,24]. However, the negative associations with breast cancer remain unclear. Patients with breast cancer may suffer from REM or NREM sleep problems [25,26]. A previous study revealed that demoralization mediates the effect of stress on sleep disturbances in patients with breast cancer [18]. An assessment of the association between demoralization and sleep quality is important. Furthermore, spiritual care for cancer patients must be interdisciplinary and culturally appropriate. Research exploring spirituality and spiritual care among cancer patients in Asian countries or globally is scarce [27]. Demoralization is associated with spiritual interests in oral cancer [23]. An assessment of demoralization may help in understanding spiritual care in breast cancer.

Depression and suicide rates are high among breast cancer survivors [28]. The increased risk of suicide in breast cancer survivors is likely underestimated, as suicide is often classified under “other causes of death”, and this may occur more often in women who have cancer. Suicide almost always occurs among individuals with mental health disorders, particularly depression [29]. Extant literature distinguishes between demoralization and depression; however, the two are strongly correlated in cancer patients [17,30,31,32,33,34]. Depression and demoralization are both high-risk factors for suicide in cancer patients, and patients with both are at higher risk [23,24,35]. Demoralization is independently associated with suicide after controlling for the severity of depression [36]. Nevertheless, the association between demoralization and depression in patients with breast cancer remains understudied.

### 1.3. The Present Study

There are several research gaps concerning the demoralization of breast cancer. First, previous studies did not confirm the severity and prevalence of demoralization in patients with breast cancer. Second, demoralization is a psychological response of patients with breast cancer; its associations with quality of life, sleep quality, and spiritual interests are not well understood. Third, the topic of whether quality of life, sleep quality, and spiritual interests can predict demoralization in patients with breast cancer has not been determined. Fourth, depression and its association with demoralization, quality of life, sleep quality, spiritual interest, and suicide in patients with highly demoralized breast cancer remain unclear. To evaluate these issues in inpatients settings, this cross-sectional study assessed demoralization in breast cancer inpatients and its relationship with demographic factors, quality of life, sleep quality, spiritual fulfillment, and suicide risk. Additionally, the study analyzed the differences between depressed and non-depressed patients with demoralization.

## 2. Materials and Methods

### 2.1. Participants

Convenience sampling was used to recruit participants (*N* = 121). Researchers expected to collect more measures in the inpatient settings. Breast cancer patients hospitalized in medical care for surgery, chemotherapy, symptom relief, and radiotherapy at Taichung Veterans General Hospital (Taichung, Taiwan) from August to December 2019 were invited to participate. The inclusion criteria were a diagnosis of breast cancer, age ≥20 years, and the ability to communicate in Mandarin.

### 2.2. Measures

Trained research assistants administered the following surveys and assessed relevant sociodemographic factors (age, sex, education, marital status, religious affiliation, employment status, and monthly income), cancer status (stage, location, and TNBC), treatment in current hospitalization, and time since diagnosis.

#### 2.2.1. Mandarin Version of Demoralization Scale (DS-MV)

DS-MV is a demoralization scale developed by Kissane [16]. It spans five distinct dimensions: loss of meaning (five items), disheartenment (six items), dysphoria (five items), sense of failure (four items), and helplessness (four items). The DS-MV demonstrated high reliability (overall Cronbach’s alpha = 0.928; subscale Cronbach’s alpha range = 0.63–0.85). Research has also shown a positive correlation (γ = 0.703, *p* < 0.001) between DS-MV and Beck Hopelessness Scale scores, and a negative correlation (γ = −0.680, *p* < 0.001) between DS-MV and McGill Quality of Life Questionnaire scores. DS-MV scores greater than 30 indicate high demoralization [15,16].

#### 2.2.2. Chinese Version of European Organization for Research and Treatment of Cancer Quality of Life Questionnaire (EORTC QLQ-C30)

The European Organization for Research and Treatment of Cancer Quality of Life Questionnaire (EORTC QLQ-C30) has been translated and validated in over 100 languages and used in numerous studies worldwide. We obtained the Chinese version of the EORTC QLQ-C30 in Taiwan with an agreement license from the Quality of Life Unit of the EORTC Data Center. The questionnaire contained 30 items evaluating general health, with specific physical, emotional, and social domains. Items were grouped into five functional scales: physical, role, cognitive, emotional, and social. It also has three symptom scales (fatigue, nausea, and pain), as well as six individual items evaluating the intensity of the following symptoms: dyspnea, sleeplessness, lack of appetite, constipation, diarrhea, and financial problems. The last two items constitute the overall health assessment. Items use a 4-point Likert response scale (1 = never to 4 = often) [37], with higher scores indicating higher levels of the given construct. A high global health status represents a high quality of life, and a high symptom score represents a high level of symptoms.

#### 2.2.3. Chinese Version of the Pittsburgh Sleep Quality Index (CPSQI)

The Pittsburgh Sleep Quality Index (PSQI) evaluates subjective sleep quality. There are 19 items assessing sleep patterns and quality over the past month. Four items concern sleep timing, with multiple-choice questions based on sleep quality. The 19 items span seven dimensions: sleep quality, habitual sleep efficiency, sleep latency, daytime dysfunction, sleep duration, sleep disturbances, and the use of sleep medications. The sum of the responses on these seven components constitutes a global score ranging from 0 to 21, with lower scores indicating better sleep quality. Scores greater than 5 indicate possible sleep pathology [38]. Subsequently, the CPSQI was developed and properly translated, becoming a reliable and valid tool to screen for sleep pathology. It is a useful self-administered tool with high sensitivity. Scores greater than 5 on the CPSQI yield 98% sensitivity and 55% specificity [39].

#### 2.2.4. Spiritual Interests Related to Illness Tool Chinese Version (C-SpIRIT)

Taylor developed the 44-item SpIRIT to measure the fulfillment of spiritual interests among cancer patients and their families [40]. It spans eight dimensions: possessing a positive perspective, having a relationship with God, giving love to others, receiving love from others, revaluating beliefs, seeking meaning in life, practicing religion, and preparing for death. Items use a 5-point Likert response scale, with higher scores indicating stronger spiritual interests in a specific category. Lin [41] modified the Chinese version of the scale for Taiwanese patients. This version contains 21 items spanning five dimensions: beliefs/religion, positive attitude toward life, love to/from others, seeking the meaning of life, and a peaceful mind.

#### 2.2.5. Patient Health Questionnaire (PHQ-9)

The PHQ-9 is a nine-item instrument administered to patients in primary care settings to screen for depression (presence and severity). This can be used to establish a diagnosis of depression according to the DSM-5 criteria. Test–retest reliability was assessed by computing the correlation (0.84) between the PHQ-9 scores obtained from in-person and phone interviews with the same patients [42]. In an assessment of construct validity, the correlation between the scores on the PHQ-9 and SF-20 mental health scales was 0.73. To assess criterion validity, a mental health professional validated depression diagnoses using PHQ-9 scores from 580 participants, resulting in 88% sensitivity and 88% specificity [42].

#### 2.2.6. Questionnaire on Suicide Risk

The DS-MV and PHQ-9 define suicide risk in distinct ways. First, items from the DS-MV that indicate suicide risk include items 14 (“Life is no longer worth living”) and 20 (“I would rather not be alive”). Answering “yes” to at least one of these statements indicated a higher risk of suicide. Second, responses greater than 0 on item 9 of the PHQ-9 (“Thoughts that you would be better off dead, or thoughts of hurting yourself in some way?”) indicated a higher risk of suicide.

### 2.3. Data Analysis

We divided participants into high- and low-demoralization groups based on the DS-MV cutoff score of 30 [15,16]. We used the receiver operating characteristic (ROC) curve classification to demoralize the associations between quality of life and spiritual interests. Furthermore, the high-demoralization group completed the PHQ-9, and was subdivided into depressed and non-depressed groups based on a cutoff score of 10 [42]. Sociodemographic and cancer status data were expressed as frequencies and percentages. We evaluated between-group differences using chi-square tests, Fisher’s exact tests, Mann-Whitney tests, and Spearman’s rank correlation coefficients. We then estimated the EORTC QLQ-C30, Global CPSQI, and C-SpIRIT scores associated with demoralization using logistic regression. All tests were set at a significance level of 0.05. All analyses were performed using IBM SPSS (version 22.0, IBM, New York, USA).

## 3. Results

### 3.1. Sociodemographic Characteristics and Demoralization in Breast Cancer Patients

There were 34 participants (28.10%) in the high-demoralization group and 87 (71.90%) in the low-demoralization group (Table 1). Sociodemographic and cancer status variables, such as age, tumor stage, TNBC, time since diagnosis, recurrence, marital status, religious affiliation, employment status, and income, did not differ significantly between the groups. Treatment during hospitalization did not differ significantly, except for participants hospitalized for surgery (low demoralization: 25/87 (28.74%) and high demoralization: 3/34 (8.82%) (*p* = 0.036)). Education levels were significantly different between the groups (*p* = 0.024). Those in the low-demoralization group had completed mainly senior high school (40/87) (45.98%) and college (36/87) (41.38%), while those in the high-demoralization group had completed college (19/34) (55.88%), senior high school (8/34) (23.53%), and elementary school (4/34) (11.76%).

### 3.2. Demoralization’s Relationship with Quality of Life, Sleep Quality, and Spiritual Interests

The mean DS-MV scores were 25.12 ± 14.89, 18.02 ± 9.34, and 43.29 ± 10.29 for all participants, the low-demoralization group, and the high-demoralization group, respectively. The mean EORTC QLQ-C30 scores were 60.20 ± 24.03, 65.80 ± 22.17, and 45.85 ± 22.88 for all participants, the low-demoralization group, and the high-demoralization group, respectively (*p* < 0.001). The mean CPSQI scores were 8.45 ± 4.51, 7.86 ± 4.28, and 10.06 ± 4.75 for all participants, the low-demoralization group, and the high-demoralization group, respectively (*p* = 0.022). The mean C-SpIRIT scores were 3.85 ± 0.58, 3.98 ± 0.48, and 3.50 ± 0.66 for all participants, the low-demoralization group, and the high-demoralization group, respectively (*p* < 0.001). For quality of life, high demoralization was associated with lower scores on the EORTC QLQ-C30; role, emotional, cognitive, and social functioning; fatigue; pain; insomnia; and appetite loss. High demoralization was associated with lower global CPSQI scores, poorer subjective sleep quality, higher sleep latency, and daytime dysfunction. High demoralization was associated with low spiritual fulfillment in all the domains (Table 2).

### 3.3. ROC Curve Analysis and Univariate and Multivariate Analyses of Variables Associated with Demoralization

Logistic regression and ROC curves were used to determine correlations between the overall quality of life, sleep quality, spiritual interests, and demoralization status. Scores on the EORTC QLQ-C30 ≥ 62.5, functional scales ≥ 72.7, symptom scales ≥ 20.3, and C-SpIRIT ≥ 3.66 were associated with demoralization (Figure 1; Table 2). The results of univariate and multivariate analyses showed that the scores for C-SpIRIT (OR = 0.15, *p* < 0.001), EORTC QLQ-C30 ≥ 62.5 (OR = 0.21, *p* = 0.002), and C-SpIRIT ≥ 3.66 (OR = 0.11, *p* < 0.001) were strongly associated with demoralization (Table 3).

### 3.4. Depression in the High-Demoralization Group

There were 15/34 (44.12%) participants in the high-demoralization group with PHQ-9 scores of ≥ 10. Sociodemographic and cancer status variables, such as age, tumor stage, TNBC, treatment, recurrence, marital status, religious affiliation, employment status, and income, did not differ significantly between the subgroups. The time since diagnosis was significantly different between the depression and non-depression subgroups (*p* = 0.040). The time since diagnosis was 3 months to 1 year (10/19) (52.63%) in the non-depression subgroup and <3 months (9/15) (60.00%) or 3 months to 1 year (5/15) (33.33%) in the depression subgroup. Education level was significantly different between the depression and non-depression subgroups (*p* = 0.042). The majority of those in the non-depression group had completed college (14/19) (73.68%), while those in the depression group had completed senior high school (6/15) (40.00%) and college (5/15) (33.33%). Work (part-time or full-time) was significantly different between the depression (46.67%) and non-depression (84.21%) subgroups (*p* = 0.030) (Table 4).

The mean DS-MV scores were 48.20 ± 11.43 and 39.42 ± 7.52 for the depression and non-depression subgroups, respectively (*p* = 0.010). The mean EORTC QLQ-C30 scores were 30.53 ± 16.24 and 57.95 ± 20.11 for the depression and non-depression group subgroups, respectively (*p* < 0.001). The mean CPSQI scores were 12.27 ± 4.57 and 8.32 ± 4.22 for the depression and non-depression subgroups, respectively (*p* = 0.017). The mean C-SpIRIT scores did not differ significantly between the depression and non-depression subgroups (Table 5).

### 3.5. Suicide Risk in Breast Cancer Patients

Suicide risk did not differ significantly between the high (5.88%) and low (1.65%) demoralization groups or between the depressed (6.67%) and non-depressed (5.26%) subgroups when defining suicide risk per responses to items 14 and 20 on the DS-MV; it was also not significantly different between the depressed (20.00%) and non-depressed (10.52%) subgroups when defining suicide risk per responses to item 9 of the PHQ-9 (Table 2 and Table 5).

## 4. Discussion

This study addresses the research gap concerning spiritual and mental healthcare in patients with breast cancer. First, demoralization is common in patients with breast cancer. High demoralization in breast cancer was associated with reduced quality of life, sleep quality, and spiritual interests. Reduced quality of life and spiritual interests might predict demoralization in patients with breast cancer, and strategies that improve quality of life and spiritual interests may prevent demoralization. We found that depression is common in highly demoralized breast cancer patients. Demoralized breast cancer patients with depression may have the worst quality of life and sleep quality. Depression was not associated with spiritual interests in demoralized breast cancer patients; thus, improving depression may not improve spiritual care in highly demoralized patients. Therefore, specific treatment that focuses on demoralization is required. Furthermore, demoralized patients with depression may have a higher suicide risk than non-depressed patients.

In this study, the educational level was associated with demoralization. Previous studies have shown that more educated women may experience higher levels of distress owing to increased knowledge about their disease and associated therapies [43]. However, they may also have more opportunities to learn coping strategies and receive psychosocial support and treatment resources. Past research has also shown that providing appropriate educational models can improve palliative care for cancer patients [44], and that education, monthly income, and family support are protective factors for demoralization [45]. Clinicians must provide straightforward and specific coping strategies, and psychotherapy for less educated patients.

In this study, demoralization was not associated with TNBC or the tumor stage. We found that demoralization was more significantly associated with mental health, spiritual interests, and quality of life than biological factors, such as TNBC or tumor stage. Although TNBC is not associated with demoralization, it has a particularly poor prognosis [22]. Future studies should explore demoralization in patients with TNBC. In this study, participants hospitalized for surgery were, on average, less demoralized. According to a recent study, demoralization fluctuates during the treatment period [46]. Participants may perceive surgery as preferable to chemotherapy/radiotherapy, which causes long-term side effects and complications. The treatment course and cancer stage are important factors in demoralization. Patients with cancer at end-of-life stages may have higher levels of demoralization [47]. However, a longitudinal study is required in this field.

### 4.1. Demoralization and Quality of Life, Sleep Quality, and Spiritual Interests

This study found that demoralized breast cancer patients may have reduced quality of life, sleep quality, and spiritual interests. The results of this study are consistent with those of a Taiwanese study on the prevalence and severity of demoralization in breast cancer patients [18]. Demoralization is a poor prognostic factor for breast cancer care; this finding is consistent with those of a similar study on oral cancer [23], although patients with oral cancer had a higher prevalence and severity of demoralization than patients with breast cancer [17,23]. The prevalence and severity of demoralization differ among cancer types. Clinicians should provide individual assessments and psychosocial interventions tailored to different cancer patients.

This study suggests that spiritual interests and quality of life are strong predictors of demoralization in patients with breast cancer. Research on spiritual care of patients with breast cancer is limited. Further evaluation of demoralization may lead to a deeper understanding of spiritual care for breast cancer. Demoralization should be examined in a specific population of patients with breast cancer. We found that demoralization was associated with poor sleep quality, specifically subjective sleep quality, sleep latency, and daytime dysfunction. However, sleep disturbances were not a strong predictor of demoralization in multivariate analyses. Previous studies have shown that demoralization mediates the relationship between stress, sleep disturbances, and mental health [18]. However, the relationship between sleep quality and demoralization remains unclear. Demoralization and its causal or bidirectional relationship with quality of life, sleep quality, and spiritual interests should be established in future studies. Furthermore, demoralization may be a prognostic factor or predictor of quality of cancer care because of its association with indicators of holistic cancer care.

### 4.2. Quality of Life and Sleep Quality in Depressed, Demoralized Breast Cancer Patients

In this study, demoralized patients with depression had poorer quality of life and sleep quality, and 60% of the patients with depression had received their diagnosis within the last 3 months. Therefore, clinicians must assess and treat depression at an early stage. We confirmed that the proportion of employed individuals in the depression subgroup was lower than that in the non-depressed subgroup. An impaired ability to work is another important issue in cancer care. In a study by Kim et al., nearly one-third of participants reported that their work conditions had changed following cancer treatment; in the depressed group, the prevalence of decreased productivity, which is associated with depression, demoralization, and anxiety, was four times that in the non-depressed group [48]. Additionally, depression was associated with higher PSQI scores (poorer subjective sleep quality) among highly demoralized patients. These results align with those of a previous longitudinal study on depression as a predictor of sleep disturbance in women with metastatic breast cancer [49]. Depressed and demoralized patients experienced a more severe loss of meaning and helplessness, and had the worst quality of life compared to non-depressed patients. Depression was not significantly associated with spiritual interests. Our research highlights the need for spiritual care in breast cancer patients with high levels of demoralization. Treating depression alone does not constitute adequate holistic care.

### 4.3. Suicide Risk in Depressed, Demoralized Patients

Suicide risk, as defined by items 14 and 20 of the DS-MV, was relatively lower among all our participants than among oral cancer patients [23]. A previous cohort study showed that increased suicide risk was particularly prominent in cancer patients with poor prognoses [50]. Although breast cancer is associated with a lower suicide rate than other cancers, suicide risk increases as the stage of cancer increases [51]. We found that, as defined by item 9 of the PHQ-9, 20% of the depressed subgroup of highly demoralized patients was at risk of suicide. This subgroup also had a worse quality of life and sleep quality than the non-depressed subgroup. The risk of suicide may be underestimated in patients with breast cancer. Depressed and demoralized patients have a higher suicide risk than non-depressed patients. The combination of demoralization and depression is a poor prognostic factor for patients with breast cancer. Clinicians must carefully assess the suicide risk, demoralization, and depression in this population.

### 4.4. Limitations

First, this study was cross-sectional and based on inpatients from a single institute in Taiwan, and the results may only be applied to the population of inpatients breast cancer patients. Further large-scale, longitudinal, and multicenter studies are needed to clarify these findings. Second, other physical or psychiatric diseases, treatment history, family history, treatment methods, side effects, and other relevant information concerning patients must be evaluated to improve the quality of the research. Third, this study only assessed depression in the high-demoralization group. The comorbidities of demoralization and depression in patients with breast cancer require further study. Fourth, the reliability and validity of the DS-MV and PHQ-9 in assessing suicide risk have not yet been established, which may affect the interpretability of the results. Standard suicide assessment tools, such as the Beck Scale for Suicide Ideation, should be considered in future studies. In addition, although both men and women could participate in the study, no male participants participated, and the results may only be applied to the population of female breast cancer patients. 

## 5. Conclusions

This study addressed a research gap in the field of holistic breast cancer care, and provided evidence that demoralization in breast cancer is associated with reduced quality of life, sleep quality, and spiritual interests. Quality of life and spiritual interests could be predictors of demoralization. In this study, highly demoralized breast cancer patients with depression had the worst quality of life and sleep quality. Breast cancer patients with demoralization had higher suicidal ideation (2.94–5.88%), although the difference was not significant. In participants with high demoralization and depression, suicidal ideation increased to 6.67–20.00%. In the future, demoralization and its causal or bidirectional relationship with quality of life, sleep quality, and spiritual interests should be established. The DS-MV is a self-report questionnaire that can be used as a routine screening tool to predict the quality of life, suicidal ideation, and spiritual interests among cancer patients. Therefore, further large-scale longitudinal studies are warranted.

## Figures and Tables

**Figure 1 ijerph-19-12815-f001:**
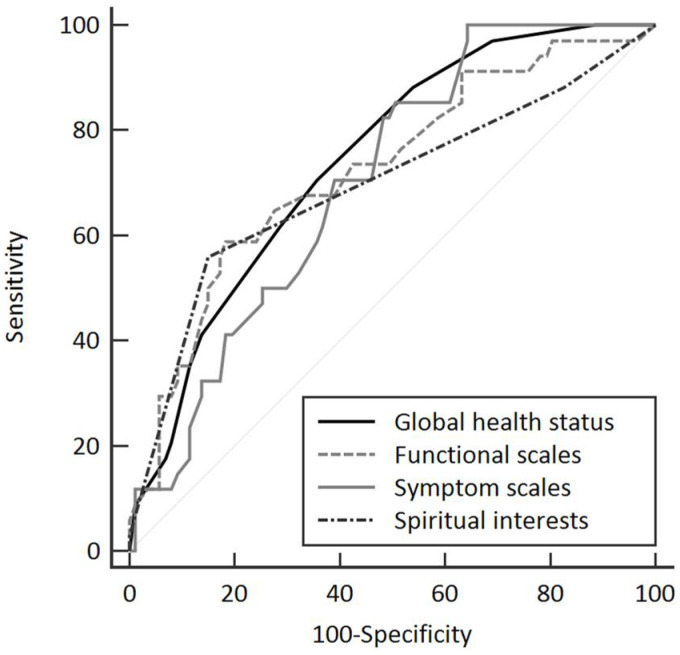
Receiver operating characteristic curve analysis of the scores of European Organization for Research and Treatment of Cancer Quality of Life Questionnaire (EORTC QLQ-C30) and spiritual interests (total scores of Spiritual Interests Related to Illness Tool Chinese Version) for demoralized patients. Global health status is the total scores of EORTC QLQ-C30.

**Table 1 ijerph-19-12815-t001:** The associations between sociodemographic characteristics and demoralization.

	Total (*n* = 121)		Low Demoralization (*n* = 87)		High Demoralization (*n* = 34)		*p* Value
Age	50.84 ± 8.59		51.10 ± 8.23		50.19 ± 9.53		0.478
Tumor Stage							0.241
0	4	(3.31%)	3	(3.45%)	1	(2.94%)	
I	26	(21.49%)	18	(20.69%)	8	(23.53%)	
II	50	(41.32%)	37	(42.53%)	13	(38.24%)	
III	23	(19.01%)	13	(14.94%)	10	(29.41%)	
IV	18	(14.88%)	16	(18.39%)	2	(5.88%)	
TNBC(Yes) ^f^	12	(9.92%)	8	(9.20%)	4	(11.76%)	0.738
Treatment							
Surgery	28	(23.14%)	25	(28.74%)	3	(8.82%)	0.036 *
Chemotherapy	91	(75.21%)	61	(70.11%)	30	(88.24%)	0.066
Symptom relief ^f^	3	(2.48%)	2	(2.30%)	1	(2.94%)	1.000
Radiotherapy ^f^	1	(0.83%)	1	(1.15%)	0	(0%)	1.000
Time since diagnosis							0.770
<3months	44	(36.36%)	32	(36.78%)	12	(35.29%)	
3 months–1 year	46	(38.02%)	31	(35.63%)	15	(44.12%)	
1–2 years	11	(9.09%)	8	(9.20%)	3	(8.82%)	
>2 years	20	(16.53%)	16	(18.39%)	4	(11.76%)	
Recurrence (Yes) ^f^	15	(12.40%)	13	(14.94%)	2	(5.88%)	0.229
Education levels							0.024 *
Elementary school	5	(4.13%)	1	(1.15%)	4	(11.76%)	
Junior school	10	(8.26%)	8	(9.20%)	2	(5.88%)	
Senior high school	48	(39.67%)	40	(45.98%)	8	(23.53%)	
College	55	(45.45%)	36	(41.38%)	19	(55.88%)	
Research institute	3	(2.48%)	2	(2.30%)	1	(2.94%)	
Marital status–married ^f^	81	(66.94%)	56	(64.37%)	25	(73.53%)	0.454
Religious affiliation							0.716
Atheist	16	(13.22%)	11	(12.64%)	5	(14.71%)	
Taoism	47	(38.84%)	34	(39.08%)	13	(38.24%)	
Buddhism	36	(29.75%)	26	(29.89%)	10	(29.41%)	
Christianity	10	(8.26%)	8	(9.20%)	2	(5.88%)	
Taoism/Buddhism	6	(4.96%)	5	(5.75%)	1	(2.94%)	
Taoism/Buddhism/Christianity	1	(0.83%)	1	(1.15%)	0	(0%)	
Others	5	(4.13%)	2	(2.30%)	3	(8.82%)	
Employment status							0.053
Unemployment	2	(1.65%)	0	(0%)	2	(5.88%)	
Retired	7	(5.79%)	5	(5.75%)	2	(5.88%)	
Housewife/househusband	27	(22.31%)	20	(22.99%)	7	(20.59%)	
Part-time job	10	(8.26%)	10	(11.49%)	0	(0%)	
Full-time job	75	(61.98%)	52	(59.77%)	23	(67.65%)	
Part-time or full time job	85	(70.25%)	62	(71.26%)	23	(67.65%)	0.865
Monthly income(USD)							0.503
USD 670 or less	25	(20.66%)	21	(24.14%)	4	(11.76%)	
USD 671–1330	54	(44.63%)	37	(42.53%)	17	(50.00%)	
USD 1331–2000	27	(22.31%)	19	(21.84%)	8	(23.53%)	
USD 2001 or more	15	(12.40%)	10	(11.49%)	5	(14.71%)	

Triple-negative breast cancer, TNBC. United States dollar, USD. ^f^ Fisher’s exact test. Chi-square test. Mann–Whitney U test. * *p* < 0.05. Continuous data were expressed as mean ± SD. Categorical data were expressed as number and percentage.

**Table 2 ijerph-19-12815-t002:** The associations between variables of rating scales and demoralization.

	Total (*n* = 121)	Low Demoralization (*n* = 87)	High Demoralization (*n* = 34)	*p* Value
Total scores of DS-MV	25.12 ± 14.89	18.02 ± 9.34	43.29 ± 10.29	<0.001 **
Loss of meaning	0.73 ± 0.64	0.47 ± 0.43	1.38 ± 0.64	<0.001 **
Dysphoria	1.24 ± 0.75	0.91 ± 0.51	2.11 ± 0.56	<0.001 **
Disheartenment	1.16 ± 0.84	0.76 ± 0.50	2.17 ± 0.65	<0.001 **
Helplessness	0.92 ± 0.72	0.64 ± 0.50	1.64 ± 0.71	<0.001 **
Sense of failure	1.16 ± 0.58	0.96 ± 0.50	1.68 ± 0.42	<0.001 **
Total scores of EORTC QLQ-C30	60.20 ± 24.03	65.80 ± 22.17	45.85 ± 22.88	<0.001 **
Functional scales	75.00 ± 15.24	78.43 ± 13.40	66.22 ± 16.29	<0.001 **
Physical functioning	81.45 ± 18.23	82.22 ± 19.25	79.47 ± 15.39	0.107
Role functioning	76.45 ± 25.12	80.46 ± 23.43	66.21 ± 26.73	0.002 **
Emotional functioning	74.63 ± 19.04	80.20 ± 14.90	60.38 ± 21.17	<0.001 **
Cognitive functioning	75.21 ± 21.06	78.54 ± 19.45	66.71 ± 22.86	0.005 **
Social functioning	67.26 ± 22.48	70.76 ± 19.69	58.32 ± 26.69	0.017 *
Symptom scales	24.95 ± 14.68	22.13 ± 14.21	32.18 ± 13.53	0.001 **
Fatigue	36.02 ± 19.66	32.78 ± 19.15	44.32 ± 18.77	0.001 **
Nausea and vomiting	13.65 ± 19.45	10.95 ± 16.20	20.56 ± 24.96	0.101
Pain	26.26 ± 22.19	22.78 ± 20.78	35.15 ± 23.47	0.006 **
Dyspnoea	17.82 ± 21.46	15.63 ± 20.76	23.41 ± 22.53	0.068
Insomnia	36.24 ± 27.64	32.05 ± 26.68	46.97 ± 27.52	0.005 **
Appetite loss	24.64 ± 22.56	20.54 ± 21.04	35.15 ± 23.23	0.001 **
Constipation	22.20 ± 24.48	19.79 ± 21.79	28.35 ± 29.78	0.192
Diarrhoea	18.35 ± 22.71	17.13 ± 22.59	21.47 ± 23.05	0.286
Financial difficulties	29.37 ± 28.32	27.48 ± 29.31	34.21 ± 25.37	0.111
Total scores of CPSQI	8.48 ± 4.51	7.86 ± 4.28	10.06 ± 4.75	0.022 *
Total scores of CPSQI>5 ^f^	76 (62.81%)	51 (58.62%)	25 (73.53%)	0.188
Subjective sleep quality	1.43 ± 0.75	1.32 ± 0.67	1.71 ± 0.87	0.015 *
Sleep latency	1.45 ± 1.05	1.29 ± 1.01	1.88 ± 1.04	0.005 **
Sleep duration	1.43 ± 1.02	1.44 ± 0.96	1.41 ± 1.18	0.909
Sleep efficiency	1.01 ± 1.19	0.93 ± 1.17	1.21 ± 1.23	0.221
Sleep disturbance	1.50 ± 0.56	1.45 ± 0.57	1.65 ± 0.54	0.058
Use of sleep medication	0.80 ± 1.26	0.70 ± 1.19	1.06 ± 1.39	0.125
Daytime dysfunction	0.85 ± 0.67	0.74 ± 0.64	1.15 ± 0.66	0.003 **
Total scores of C-SpIRIT	3.85 ± 0.58	3.98 ± 0.48	3.50 ± 0.66	<0.001 **
Related to beliefs	3.23 ± 0.84	3.35 ± 0.75	2.94 ± 0.98	0.014 *
Religion, positive attitudes toward life	4.01 ± 0.64	4.19 ± 0.55	3.54 ± 0.64	<0.001 **
Love to/from others	3.87 ± 0.72	4.02 ± 0.60	3.50 ± 0.86	0.001 **
Seeking for the meaning of life	4.01 ± 0.66	4.12 ± 0.59	3.74 ± 0.75	0.007 **
Peaceful mind	4.11 ± 0.64	4.23 ± 0.56	3.79 ± 0.73	0.002 **
Suicide risk				
Q14 of DS-MV (Yes) ^f^	2 (1.65%)	0 (0%)	2 (5.88%)	0.077
Q20 of DS-MV (Yes) ^f^	1 (0.83%)	0 (0%)	1 (2.94%)	0.281
Q14 or Q20 of DS-MV (Yes) ^f^	2 (1.65%)	0 (0%)	2 (5.88%)	0.077
ROC (Cut point)				
EORTC QLQ-C30 ≥ 62.5	66 (54.5%)	56 (64.4%)	10 (29.4%)	0.001 **
Functional scales ≥ 72.7	78 (64.5%)	66 (75.9%)	12 (35.3%)	<0.001 **
Symptom scales ≥ 20.3	72 (59.5%)	43 (49.4%)	29 (85.3%)	0.001 **
C-SpIRIT ≥ 3.66	78 (64.5%)	68 (78.2%)	10 (29.4%)	<0.001 **

Demoralization Scale Mandarin Version, DS-MV. European Organization for Research and Treatment of Cancer Quality of Life Questionnaire, EORTC QLQ-C30. Chinese Version of the Pittsburgh Sleep Quality Index, CPSQI. Spiritual Interests Related to Illness Tool Chinese Version, C-SpIRIT. Receiver operating characteristic, ROC. ^f^ Fisher’s exact test. Chi-square test. Mann–Whitney U test. * *p* < 0.05, ** *p* < 0.01. Continuous data were expressed as mean ± SD. Categorical data were expressed as number and percentage.

**Table 3 ijerph-19-12815-t003:** Univariate and multivariate analyses of demoralization in all patients.

	Univariate			Multivariate			Multivariate		
	OR	95% CI	*p* Value	OR	95% CI	*p* Value	OR	95% CI	*p* Value
EORTC QLQ-C30	0.96	(0.95–0.98)	<0.001 **	0.97	(0.95–0.99)	0.003 **			
Global CPSQI	1.11	(1.02–1.22)	0.018 *	1.11	(0.99–1.25)	0.069			
Global CPSQI > 5	1.87	(0.78–4.48)	0.161						
C-SpIRIT	0.17	(0.07–0.43)	<0.001 **	0.15	(0.05–0.41)	<0.001 **			
ROC (Cut point)									
EORTC QLQ-C30 ≥ 62.5	0.23	(0.10–0.54)	0.001 **				0.21	(0.08–0.57)	0.002 **
C-SpIRIT ≥ 3.66	0.12	(0.05–0.29)	<0.001 **				0.11	(0.04–0.29)	<0.001 **

European Organization for Research and Treatment of Cancer Quality of Life Questionnaire, EORTC QLQ-C30. Chinese Version of the Pittsburgh Sleep Quality Index, CPSQI. Spiritual Interests Related to Illness Tool Chinese Version, C-SpIRIT. Receiver operating characteristic, ROC. Logistic regression. * *p* < 0.05, ** *p* < 0.01.

**Table 4 ijerph-19-12815-t004:** The associations between sociodemographic characteristics and depression in the high-demoralization group.

	Total (*n* = 34)	Non-Depression (*n* = 19)		Depression (*n* = 15)		*p* Value
Age	50.19 ± 9.53	49.74 ± 9.75		50.76 ± 9.56		0.700
Tumor Stage						0.619
0	1	0	(0%)	1	(6.67%)	
I	8	6	(31.58%)	2	(13.33%)	
II	13	7	(36.84%)	6	(40.00%)	
III	10	5	(26.32%)	5	(33.33%)	
IV	2	1	(5.26%)	1	(6.67%)	
TNBC (yes) ^f^	4	3	(15.79%)	1	(6.67%)	0.613
Treatment						
Surgery	3	1	(5.26%)	2	(13.33%)	0.571
Chemotherapy	30	17	(89.47%)	13	(86.67%)	1.000
Symptom relief ^f^	1	1	(5.26%)	0	(0%)	1.000
Radiotherapy ^f^	0	0	(0%)	0	(0%)	---
Time since diagnosis						0.040 *
<3months	12	3	(15.79%)	9	(60.00%)	
3 months–1 year	15	10	(52.63%)	5	(33.33%)	
1–2 years	3	3	(15.79%)	0	(0%)	
>2 years	4	3	(15.79%)	1	(6.67%)	
Recurrence (Yes) ^f^	2	1	(5.26%)	1	(6.67%)	1.000
Education levels						0.042 *
Elementary school	4	3	(15.79%)	1	(6.67%)	
Junior school	2	0	(0%)	2	(13.33%)	
Senior high school	8	2	(10.53%)	6	(40.00%)	
College	19	14	(73.68%)	5	(33.33%)	
Research institute	1	0	(0%)	1	(6.67%)	
Marital status: married ^f^	25	13	(68.42%)	12	(80.00%)	0.697
Religious affiliation						0.123
Atheist	5	5	(26.32%)	0	(0%)	
Taoism	13	8	(42.11%)	5	(33.33%)	
Buddhism	10	5	(26.32%)	5	(33.33%)	
Christianity	2	0	(0%)	2	(13.33%)	
Taoism/Buddhism	1	0	(0%)	1	(6.67%)	
Taoism/Buddhism/Christianity	0	0	(0%)	0	(0%)	
Others	3	1	(5.26%)	2	(13.33%)	
Employment status						0.093
Unemployment	2	0	(0%)	2	(13.33%)	
Retired	2	1	(5.26%)	1	(6.67%)	
housewife/househusband	7	2	(10.53%)	5	(33.33%)	
Part-time job	0	0	(0%)	0	(0%)	
Full-time job	23	16	(84.21%)	7	(46.67%)	
Part-time or full time job ^f^	23	16	(84.21%)	7	(46.67%)	0.030 *
Monthly income(USD)						0.147
USD 670 or less	4	1	(5.26%)	3	(20.00%)	
USD 671–1330	17	8	(42.11%)	9	(60.00%)	
USD 1331–2000	8	7	(36.84%)	1	(6.67%)	
USD 2001 or more	5	3	(15.79%)	2	(13.33%)	

Chi-square test. ^f^ Fisher’s exact test. Mann-Whitney U test.* *p* < 0.05. Continuous data were expressed as mean ± SD. Categorical data were expressed as number and percentage. Triple-negative breast cancer, TNBC. United States dollar, US.

**Table 5 ijerph-19-12815-t005:** The associations between the variables of rating scales and depression in the high-demoralization group.

	Total (*n* = 34)	Non-Depression (*n* = 19)	Depression (*n* = 15)	*p* Value
Total scores of DS-MV	43.29 ± 10.29	39.42 ± 7.52	48.20 ± 11.43	0.010 *
Loss of meaning	1.38 ± 0.64	1.16 ± 0.51	1.67 ± 0.69	0.041 *
Dysphoria	2.11 ± 0.56	2.03 ± 0.57	2.21 ± 0.54	0.373
Disheartenment	2.17 ± 0.65	2.00 ± 0.69	2.39 ± 0.53	0.073
Helplessness	1.64 ± 0.71	1.34 ± 0.44	2.02 ± 0.82	0.015 *
Sense of failure	1.68 ± 0.42	1.62 ± 0.39	1.76 ± 0.45	0.461
Total scores of EORTC QLQ-C30	45.85 ± 22.88	57.95 ± 20.11	30.53 ± 16.24	<0.001 **
Functional scales	66.22 ± 16.29	73.92 ± 13.06	56.47 ± 14.98	0.001 **
Physical functioning	79.47 ± 15.39	82.89 ± 14.69	75.13 ± 15.65	0.065
Role functioning	66.21 ± 26.73	68.53 ± 27.11	63.27 ± 26.88	0.631
Emotional functioning	60.38 ± 21.17	74.26 ± 12.71	42.80 ± 16.00	<0.001 **
Cognitive functioning	66.71 ± 22.86	77.21 ± 15.79	53.40 ± 23.91	0.001 **
Social functioning	58.32 ± 26.69	66.68 ± 24.82	47.73 ± 25.93	0.025 *
Symptom scales	32.18 ± 13.53	25.02 ± 8.80	41.24 ± 13.21	<0.001 **
Fatigue	44.32 ± 18.77	36.05 ± 14.86	54.80 ± 18.33	0.004 **
Nausea and vomiting	20.56 ± 24.96	11.37 ± 15.70	32.20 ± 29.87	0.043 *
Pain	35.15 ± 23.47	27.95 ± 22.87	44.27 ± 21.60	0.031 *
Dyspnoea	23.41 ± 22.53	19.11 ± 16.74	28.87 ± 27.91	0.370
Insomnia	46.97 ± 27.52	34.95 ± 20.84	62.20 ± 27.96	0.006 **
Appetite loss	35.15 ± 23.23	29.53 ± 10.40	42.27 ± 32.17	0.148
Constipation	28.35 ± 29.78	22.68 ± 22.36	35.53 ± 36.73	0.425
Diarrhoea	21.47 ± 23.05	19.16 ± 20.16	24.40 ± 26.70	0.713
Financial difficulties	34.21 ± 25.37	24.42 ± 21.78	46.60 ± 24.76	0.011 *
Total scores of CPSQI	10.06 ± 4.75	8.32 ± 4.22	12.27 ± 4.57	0.017 *
Total scores of CPSQI > 5 ^f^	25 (73.53%)	12 (63.16%)	13 (86.67%)	0.240
Subjective sleep quality	1.71 ± 0.87	1.32 ± 0.82	2.20 ± 0.68	0.003 **
Sleep latency	1.88 ± 1.04	1.68 ± 1.16	2.13 ± 0.83	0.290
Sleep duration	1.41 ± 1.18	1.16 ± 1.07	1.73 ± 1.28	0.172
Sleep efficiency	1.21 ± 1.23	0.95 ± 1.13	1.53 ± 1.30	0.203
Sleep disturbance	1.65 ± 0.54	1.47 ± 0.51	1.87 ± 0.52	0.054
Use of sleep medication	1.06 ± 1.39	0.79 ± 1.36	1.40 ± 1.40	0.103
Daytime dysfunction	1.15 ± 0.66	0.95 ± 0.62	1.40 ± 0.63	0.066
Total scores of C-SpIRIT	3.50 ± 0.66	3.43 ± 0.63	3.59 ± 0.71	0.925
Related to beliefs	2.94 ± 0.98	2.77 ± 0.92	3.15 ± 1.05	0.366
Religion, positive attitudes toward life	3.54 ± 0.64	3.43 ± 0.61	3.67 ± 0.67	0.187
Love to/from others	3.50 ± 0.86	3.41 ± 0.85	3.62 ± 0.90	0.829
Seeking for the meaning of life	3.74 ± 0.75	3.70 ± 0.66	3.78 ± 0.87	0.868
Peaceful mind	3.79 ± 0.73	3.82 ± 0.73	3.76 ± 0.76	0.594
Suicide risk				
Q14 of DS-MV (Yes) ^f^	2 (5.88%)	1 (5.26%)	1 (6.67%)	1.000
Q20 of DS-MV (Yes) ^f^	1 (2.94%)	0 (0%)	1 (6.67%)	0.441
Q14 or Q20 of DS-MV (yes)^f^	2 (5.88%)	1 (5.26%)	1 (6.67%)	1.000
Q9 of PHQ-9 > 0 (Yes) ^f^	5 (14.71%)	2 (10.52%)	3 (20.00%)	0.634

Demoralization Scale Mandarin Version, DS-MV. Patient Health Questionnaire -9, PHQ-9. European Organization for Research and Treatment of Cancer Quality of Life Questionnaire, EORTC QLQ-C30. Chinese Version of the Pittsburgh Sleep Quality Index, CPSQI. Spiritual Interests Related to Illness Tool Chinese Version, C-SpIRIT. ^f^ Fisher’s exact test. Chi-square test. Mann–Whitney U test. * *p* < 0.05, ** *p* < 0.01. Continuous data were expressed as mean ± SD. Categorical data were expressed as number and percentage.

## Data Availability

Not applicable.

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
