# Peer review of "Demoralization and Its Association with Quality of Life, Sleep Quality, Spiritual Interests, and Suicide Risk in Breast Cancer Inpatients: A Cross-Sectional Study"

_ijerph, 2022, doi:10.3390/ijerph191912815_

Round 1
Reviewer 1 Report
In the conclusions, I would suggest a stronger emphasis on the examined link between demoralization and suicide
Reviewer 2 Report
The manuscript discussed demoralization and its association with quality of life and other mental health conditions in patients with breast cancer in a signal inpatient setting. The study is interesting; however, I have several suggestions and concerns about the manuscript.
As a reader, the introduction section could be reorganized to better guide readers to understand:
a. why would you like to focus on patients with "breast cancers?" E.g., breast cancer is the most prevalent cancer type among women/men or in XX years or in Taiwan, etc. Any more updated statistics for this cancer?
b. what relationships between demoralization and (breast) cancer? I would suggest considering combing 1.1 and 1.2 succinctly. The current sequences of sentences are confusing and some sentences seem off to me (e.g., lines 67-68, 72-73, etc.)
It seems that there is a typo in the title of 1.3 (TDdemoralization). I believe it should be Demogralization.
For 1.4, you mentioned four gaps and your study would address them. However, I am not quite sure that the current study would be able to answer these four questions.
For the Materials and Methods section, I am not sure if you excluded participants with previous mental health or other chronic conditions (i.e., did you have any exclusion criteria?). Did you limit your participants to only females or males? You mention that you assessed "sex" in 2.2 but this variable was not presented in tables. Also, you may want to mention why you chose "breast cancer patients hospitalized in acute care for various reasons" in 2.1. Why hospitalized patients, not patients in outpatient settings? What would "various reasons" include? Would these acute care for these reasons be related to breast cancers?
In 2.3, I would suggest citing your cut-off point of 30 in line 176 and separating logistic regression in line 181 to become another sentence as this is for modeling, not descriptive analyses as chi-square tests, etc.
For your tables, there are many cells with zero values but you had p values for the tests. I wondered if you could talk about how you conduct these comparisons. Did you do any imputations?
Round 2
Reviewer 2 Report
The authors have addressed most of the comments. The revision looks fine to me now. Thank you.
Author Response
We would like to thank you for providing your constructive and detailed review comments on our manuscript. The recommendations and advice have helped us to significantly enhance the quality of the manuscript.